# Superimposed effects of typical local circulations driven by mountainous topography and aerosol-radiation interaction on heavy haze in the Beijing-Tianjin-Hebei central and southern plains in winter

Yue Peng[1], Hong Wang[1], Xiaoye Zhang[1,2], Zhaodong Liu[1], Wenjie Zhang[1], Siting Li[1], Chen Han[1], Huizheng Che[1]

[1]State Key Laboratory of Severe Weather (LASW) & Key Laboratory of Atmospheric Chemistry of China Meteorological Administration, Chinese Academy of Meteorological Sciences (CAMS), Beijing, 100081, China
[2]Center for Excellence in Regional Atmospheric Environment, IUE, Chinese Academy of Sciences, Xiamen, 361021, China

*Correspondence to*: Hong Wang (wangh@cma.gov.cn); Xiaoye Zhang (xiaoye@cma.gov.cn)

**Abstract.** Although China's air quality has substantially improved in recent years due to the vigorous emission reduction, the Beijing-Tianjin-Hebei (BTH) region, especially its central and southern plains at the eastern foot of the Taihang Mountains, has been the most polluted area in China with persistent and severe haze in winter. Combining meteorology-chemistry coupled model simulations and multiple observations, this study explored the causes of several heavy haze events in this area in January 2017, focusing on local circulations related to mountain terrain. The study results showed that on the weather scale, the configuration of the upper, middle, and lower atmosphere provided favorable weather and water vapor transport conditions for the development of haze pollution. Under the weak weather-scale systems, local circulation played a dominant role in the regional distribution and extreme values of $PM_{2.5}$. Influenced by the Taihang and Yanshan Mountains, vertical circulations and wind convergence zone were formed between the plain and mountain slopes. The vertical distribution of pollutants strongly depended on the intensity and location of the circulation. The circulation with high intensity and low altitude was more unfavorable for vertical and horizontal diffusion of near-surface pollutants. More importantly, we found that aerosol-radiation interaction (ARI) significantly amplified the impacts of local vertical circulations on heavy haze by two mechanisms. First, ARI strengthened the vertical circulations at the lower levels, with the zonal wind speeds increasing by $0.3{\sim}0.8$ m s$^{-1}$. Meanwhile, ARI could cause a substantial downward shift of the vertical circulations (~100 m). Second, ARI weakened the horizontal diffusion of pollutants by reducing the westerly winds and enhancing the wind convergence as well as the southerly winds in the polluted area. Under these two mechanisms, pollutants could only recirculate in a limited space. This superposition of typical local circulation and ARI eventually contributed to the accumulation of pollutants and the consequent deterioration of haze pollution in the region.

# 1 Introduction

China's air quality has considerably improved in recent years because of aggressive emission reduction measures (Q. Zhang et al., 2019; Zheng et al., 2018). However, the large urban agglomeration such as the Beijing-Tianjin-Hebei (BTH) region and the Yangtze River Delta (YRD) still frequently suffer from persistent heavy haze pollution and the deterioration of atmospheric visibility that it causes, especially in winter (Huang et al., 2020; Peng et al., 2020). Particulate matter with an aerodynamic diameter of less than 2.5 μm ($PM_{2.5}$) is the primary aerosol component of haze and a significant factor affecting

visibility. During heavy haze pollution, $PM_{2.5}$ concentrations often exceeded 300 μg m$^{-3}$, and sometimes even 500 μg m$^{-3}$ in these areas (Peng et al., 2021; Wang et al., 2018; Zhang et al., 2020). Emissions and meteorological conditions are two key factors affecting pollutants. However, emissions in a region do not change much in the short term, when pollution levels may be dominated by regional or local meteorological conditions (Wang et al., 2018; Zheng et al., 2015; Zhong et al., 2017). For regional or local air pollution, the link between aerosol-radiation interaction (ARI), planetary boundary layer (PBL), and

long-range transport has been investigated in recent years. The studies illustrated that there is a positive feedback between aerosols and the PBL: During heavy haze pollution, high aerosol concentrations weaken the turbulence in the lower troposphere mainly by scattering the solar radiation, thus inhibiting the development of PBL (Miao et al., 2019; Peng et al., 2021; Quan et al., 2013; Wang et al., 2018; Zhong et al., 2018a), while absorbing aerosols, such as black carbon (BC), can heat the upper PBL, and further enhance the stability of the atmospheric stratification (Ding et al., 2016; Huang et al., 2018).

The decreased PBL then increases near-surface relative humidity (RH) by weakening the vertical transport of water vapor; the increased RH in turn promotes the formation of secondary aerosols (Li et al., 2017; Liu et al., 2018). These aerosol direct and semi-direct effects or ARI will eventually deteriorate haze pollution. Aerosols can also act as cloud condensation nuclei or ice nuclei, modifying cloud physical and radiative properties by participating in cloud microphysical processes, this aerosol-cloud interaction in turn affects the structure and development of the PBL (Zhang et al., 2015; Zhao et al., 2017).

Moreover, the PBL feedback can interact with long-range transport through ARI, and this interaction then amplifies transboundary air pollution transport between northern and eastern China (Huang et al., 2020).

In addition to the interactions above, local circulations driven by unique topography also play an important role in the variations of PBL structure as well as the spatial and temporal distribution of pollutants (Chen et al., 2009; Liu et al., 2009; Miao et al., 2015; Zhang et al., 2018). The BTH region is located in the North China Plain (NCP), with the Yanshan

Mountains to the north, Taihang Mountains to the west, and the Bohai Sea to the east (Fig. 1a). The elevation difference between these two mountains and the NCP can reach 1500–2000 m. Such a complex geographical environment makes the BTH region have unique local atmospheric circulation characteristics and is prone to local accumulation or regional transport of pollutants. Chen et al. (2009) found that the elevation of the pollution layer in Beijing is associated with the mountain-plain breeze, which causes a rapid increase of pollutants in the near-surface in this area. The intensity of local atmospheric

circulation can strongly affect the removal and accumulation of local pollutants. In the absence of strong weather systems, the well-developed valley wind circulation and sea breeze circulation over the BTH region facilitates the long-distance

transport of pollutants (Miao et al., 2015). On the contrary, weak local circulations make pollutants recirculate in a limited space and accumulate continuously (Lo et al., 2006; Sun et al., 2013). In addition, several topographic sensitivity experiments have been conducted to examine their effects on the low-level circulation and PBL structure in the BTH (Wang et al., 2019; Zhang et al., 2018), and the results highlight the significance of topography in the formation and accumulation of haze pollution. Despite the fact that there have been numerous previous studies on causes of haze pollution in the BTH, most of them concentrated on the effects of ARI or topography alone, and few studies focused on the link between ARI and topography-induced local circulation, and the possible impact of this interaction on haze pollution.

The BTH suffered several heavy haze episodes between December 2016 and February 2017, with the persistent severe pollution in January 2017 being the most representative. Therefore, using the online coupled atmospheric chemistry model GRAPES_Meso5.1/CUACE, this study focused on January 2017 and comprehensively explored the link between local circulation, ARI, and haze, especially the impacts of ARI on local circulation.

## 2 Method and data

### 2.1 GRAPES_Meso5.1/CUACE

GRAPES_Meso5.1/CUACE is an online regional atmospheric chemistry model designed for both operational and research applications. There are two major parts of this model: a weather model GRAPES_Meso5.1 and a chemistry model CUACE. The former is a mesoscale weather prediction model that primarily consists of a fully compressible non-hydrostatical model core and a modularized physics package (Chen et al., 2008); the latter is an online chemistry model that is mainly composed of aerosol and gaseous chemistry modules with emission and dynamic processes (Gong and Zhang, 2008). Wang et al. (2022) established this updated model and provided a comprehensive description of the model. In this model, Peng et al. (2022) implemented the ARI mechanism for the two-way feedback between aerosols and weather processes by incorporating the real-time calculated aerosol optical parameters.

The model domain in this study is centered over the BTH region, covering an area of 33–45 °N in latitude and 110–125 °E in longitude (Fig. 1a). The model has a horizontal resolution of 10 km and 49 unevenly spaced vertical levels ranging from near-surface to 33 km. The physical configuration options selected in this study include the Thompson microphysics (Thompson et al., 2008), the KF cumulus scheme (Kain, 2004), the RRTMG longwave radiation scheme (Mlawer et al., 1997), the Goddard shortwave radiation scheme (Chou et al., 1998) including ARI mechanism, the MRF boundary layer scheme (Hong and Pan, 1996), the MM5 surface layer scheme (Zhang and Anthes, 1982) and Noah land surface scheme (Ek et al., 2003). The chemical configuration options mainly include an emissions inventory treatment system, the Second Generation Regional Acid Deposition Model (RADM2) gas-phase chemistry (Stockwell et al., 1990), and the CUACE aerosol model (Gong and Zhang, 2008; Wang et al., 2010, 2015a, 2015b).

## 2.2 Data

Five categories of data was used in this study: The global Final analysis (FNL) data with a horizontal resolution of 0.25 ° × 0.25 ° (http://rda.ucar.edu/data/ds083.3/) provided by the National Centers for Environmental Prediction (NCEP), which was used for the meteorology initial and lateral boundary fields of the model and the analysis of large-scale circulation in upper and mid-levels; Multi-year climate average of chemical tracers used for chemistry initialization of the model (Wang et al., 2022); Monthly anthropogenic emissions derived from the Multi-resolution Emission Inventory for China (MEIC) of 2017 (Zheng et al., 2018); Hourly near-surface $PM_{2.5}$ mass concentration measured by 149 state-controlled stations provided by the China National Environmental Monitoring Center (http://www.cnemc.cn/) and 210 stations provided by the Hebei Meteorological service; Vertical meteorology data for three sounding stations in the BTH, i.e., Beijing (BJ), Tangshan (TS), and Xingtai (XT), including air pressure, temperature, and wind at 08:00 and 20:00 Beijing time (BJT) each day, measured by the L-band radiosonde system.

## 2.3 Experimental design and data analysis method

The model simulation was conducted from December 29, 2016, to January 31, 2017, with a looping time of 72 h. The first 72 h simulations were considered the spin-up period. To evaluate the impacts of ARI on $PM_{2.5}$ concentration, two numerical scenarios were performed in this study. The first is the controlling simulation (CTL) with the above model configurations and ARI; the second is the sensitive experiment (EXP) which is consistent with CTL but without considering ARI. The same analysis data and emission inventory were used for both numerical scenarios.

Multiple model performance evaluation metrics were used, including the Pearson correlation coefficient (R), the mean bias (MB), the root mean square error (RMSE), the mean fractional bias (MFB), and the mean fractional error (MFE). The equations of these metrics are available in Boylan and Russell (2006).

## 3 Results and discussion

### 3.1 Model performance

Accurate reproduction of aerosol concentration variations and the vertical structure of the atmosphere is a prerequisite for quantifying ARI (Zhang et al., 2015) as well as local circulation. Figure 1 shows the distribution of observed and simulated monthly mean $PM_{2.5}$ concentrations in January 2017. The BTH region suffered from severe haze pollution in January 2017, with its regional monthly mean observed $PM_{2.5}$ concentration reaching 130 μg m$^{-3}$. Particularly at the eastern foot of the Taihang Mountains, the central and southern plains of BTH, the $PM_{2.5}$ values exceeded 200 μg m$^{-3}$ and even 250 μg m$^{-3}$. High anthropogenic emission coupled with a stable atmosphere due to the mountainous topography leads to frequent and severe haze events in this area (Fu et al., 2014). The BTH is surrounded by the Yanshan and Taihang Mountains from north to west, and such topography is not conducive to pollutant dispersion since the mountains weaken the cold air from the north

and west and block the transport of pollutants associated with easterly and southerly winds (Gao et al., 2017; Miao et al., 2015; Quan et al., 2020; Zhong et al., 2018b). The comparison of the observed and simulated $PM_{2.5}$ results (Fig. 1c–d) shows that although both simulation scenarios reproduced the distribution of $PM_{2.5}$ concentrations, the CTL results with ARI were closer to the observations. Particularly in the most polluted central and southern BTH, the maximum $PM_{2.5}$ concentration exceeded 225 μg m$^{-3}$ in CTL while was less than 200 μg m$^{-3}$ in EXP.

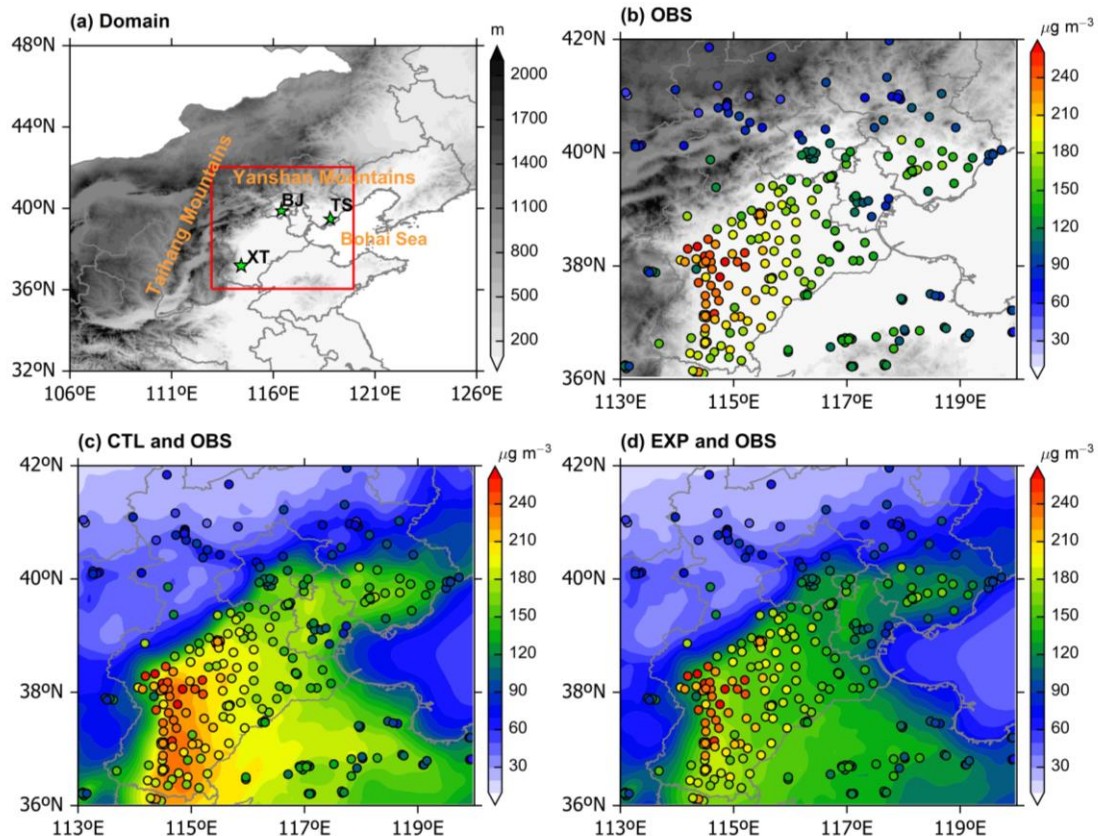

**Figure 1.** (a) Model domain (shading denotes terrain height; red rectangle shows the general location of the BTH; green starts denote the weather sounding stations) and (b–d) spatial distributions of observed (OBS: circles) and simulated (CTL and EXP: shadings) $PM_{2.5}$ concentrations in January 2017.

The hourly variation of observed and simulated $PM_{2.5}$ concentrations in the BTH was compared in Fig. 2. The model generally reproduced the temporal variation of observed $PM_{2.5}$. Compared to EXP, the model including ARI (CTL) showed better agreements with observations, with higher R (from 0.71 to 0.74), smaller MB (from -40.2 to -16.4 μg m$^{-3}$), and smaller RMSE (from 57.0 to 45.3 μg m$^{-3}$) (Table 1). Besides, both MFB and MFE showed substantial reductions, from -34.2 μg m$^{-3}$ and 37.6 μg m$^{-3}$ to -15.7 μg m$^{-3}$ and 28.5 μg m$^{-3}$, respectively (Table 1). According to model performance

goals for PM$_{2.5}$ proposed by Morris et al. (2005), both CTL and EXP simulation results achieved an average level (MFB ≤ ±60% and MFE ≤ 75%), with CTL exceeding a good level (MFB ≤ ±30% and MFE ≤ 50%) and very close to an excellent

level (MFB ≤ ±15% and MFE ≤35%). The result not only demonstrates the applicability of the model, but also the necessity of considering ARI for pollutant simulation. Moreover, the ARI effect strongly depends on the PM$_{2.5}$ concentrations (Peng et al., 2021; Zhang et al., 2022). In this study, the ARI effect was significant when the PM$_{2.5}$ concentration was larger than 100 µg m$^{-3}$ (Fig. 2), implying that the ARI effect can significantly improve the model's underestimation of the PM$_{2.5}$ peaks. Considering the essential influence of the ARI mechanism on PM$_{2.5}$ extremes and the accuracy of the simulation results,

three pollution periods were selected: January 1–7, 16–18, and 23–26. Furthermore, since the short-term characteristics of the local circulation have a greater impact on the PM$_{2.5}$ distribution than the long-term ones, it is necessary to select one day from each of the three periods to further investigate the link between local circulation and ARI and how their interaction enhances heavy haze pollution. According to the criteria of air pollution level (HJ633–2012) issued by the Ministry of Environmental Protection of China, heavy pollution is defined as the daily mean PM$_{25}$ concentration larger than 150 µg m$^{-3}$.

Therefore, January 6, 17, and 25, which reached the heavy pollution level and were in the rising stage, were finally selected as the representatives of heavy pollution days in three periods.

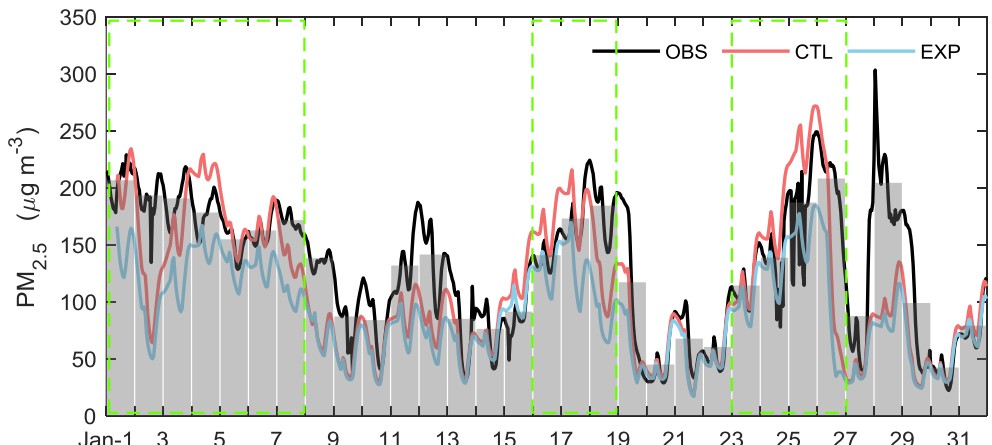

**Figure 2.** Time series of observed (OBS) and simulated (CTL and EXP) PM$_{2.5}$ concentrations in the BTH during January 2017. Grey bar: observed daily mean PM$_{2.5}$ concentrations; Green box: selected pollution period.


Table 1. Model evaluation for PM$_{2.5}$ in the BTH during January 2017.

|  | R | MB µg m$^{-3}$ | RMSE µg m$^{-3}$ | MFB % | MFE % |
|---|---|---|---|---|---|
| CTL | 0.74 | -16.4 | 45.3 | -15.7 | 28.5 |
| EXP | 0.71 | -40.2 | 57.0 | -34.2 | 37.6 |

Given the important influence of atmospheric vertical structure, especially temperature stratification, on the formation of pollutants, we further evaluated the model performance (CTL) in simulating the vertical profile of the potential temperature (PT) at BJ, TS, and XT by comparing sounding observations. As shown in Fig. 3, the model simulations reasonably reproduced the vertical distribution of temperature in BJ, TS, and XT, including a good simulation of atmospheric warming during the pollution period. For the three pollution periods (January 1–7, 16–18, and 23–26), the mean values of R for PT below 2500 m were 0.94, 0.97, and 0.97 in BJ, TS, and XT, respectively; for the heavy pollution days (January 6, 17, and 25), the R values were 0.83~0.99 in BJ, 0.96~0.99 in TS, and 0.90~0.99 in XT. The accurate representation of near-surface PM$_{2.5}$ concentrations and vertical temperature structure in the model provides a solid basis for clarifying the physical mechanisms of heavy pollution.

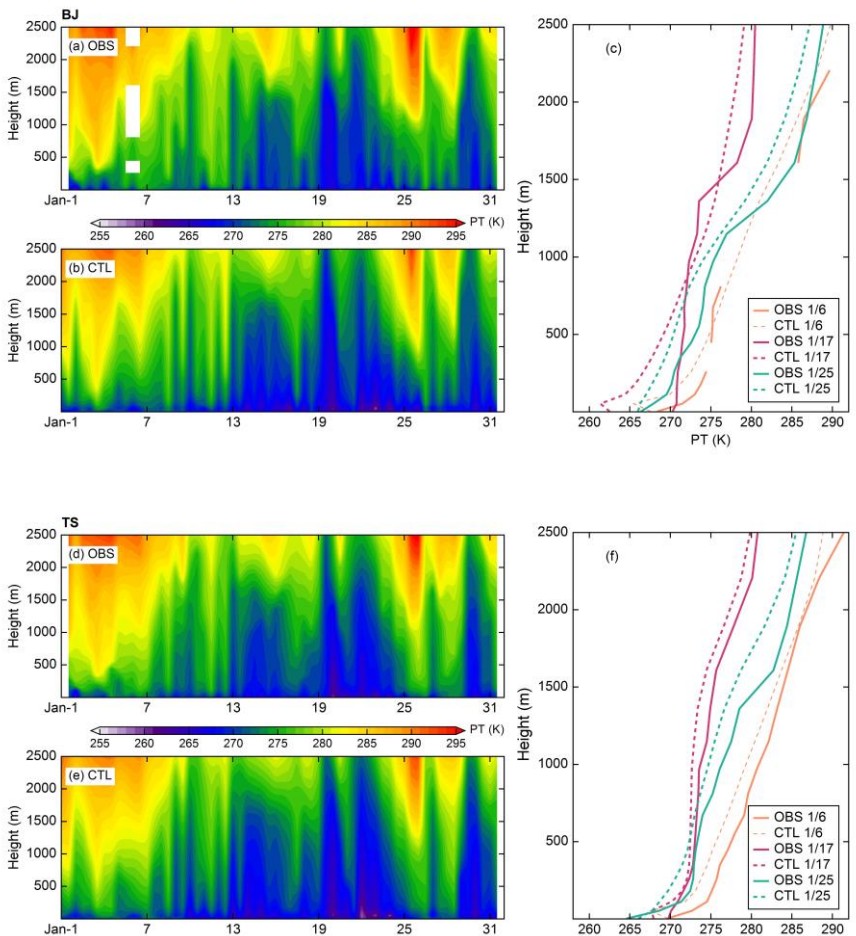

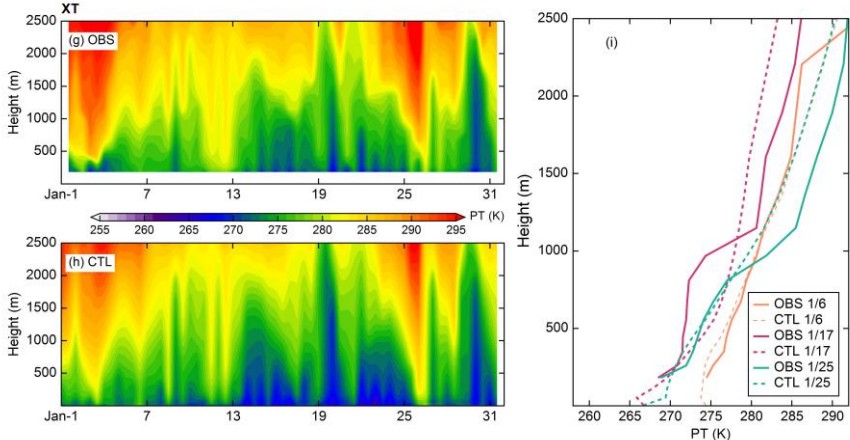


**Figure 3.** Vertical profiles of observed (OBS) and simulated (CTL) PT for (a–b) BJ, (d–e) TS, and (g–h) XT at 08:00 and 20:00 BJT in January 2017. c, f, and i denote the PT profiles for these three cities at 08:00 BJT on January 6, 17, and 25.

## 3.2 Weather situation background under haze pollution

Previous studies have shown that persistent pollution is influenced not only by the PBL and surface meteorology but also by
the configuration of upper and lower-level circulation systems (Miao et al., 2015; Wu et al., 2017). Based on the distribution characteristics of pollutants in the BTH region and the good simulation performance of the model, the horizontal distribution of the upper, middle, and lower atmosphere and surface circulation field in the pollution days was examined in this section. Before the specific discussion of the three heavy haze days, we made a general analysis of the weather situation in the three pollution periods. Fig. S1 (in the Supplement) shows the average distributions of geopotential height (GH), PT, and wind
vectors at the 500 and 700 hPa levels from the FNL data. It can be seen that the BTH region was controlled by northwesterly airflow at upper and mid-levels during the pollution periods, and its PT increased with height. This is a typical atmospheric circulation that often accompanies pollution episodes. The distributions of simulated (CTL) GH and PT at 850 hPa were similar to those at 500 and 700 hPa (Fig. S2). The BTH region was dominated by a uniform pressure field in front of the high pressure system. The weak pressure gradient resulted in weak westerly wind flows, which blocked dry and cold polar air into
this region. Under such stable atmospheric circulation conditions, the PBL development over the BTH was suppressed, which led to lower height of the PBL (PBLH) and near-surface wind speeds, contributing to the formation and maintenance of haze pollution (Fig. S3).

Based on the above results, we further compared the weather situation characteristics of the selected heavy pollution day in each period. The circulation patterns across the BTH differed considerably during these three days (Fig. 4). On January 6,
eastern China was in front of a weak north-south trough at 500 hPa, and the BTH region was controlled by southwest airflow and a slight temperature gradient. On January 17, a zonal circulation dominated East Asia, and a zonal westerly airflow was in charge of the BTH region. On January 25, mainland China was dominated by a northeast-southwest high pressure ridge,

and the BTH region was controlled by westerly and northwesterly airflow over this ridge. Moreover, the circulation patterns at 700 hPa were consistent with that at 500 hPa. All these synoptic conditions are generally considered to promote the deterioration of pollution since they impede the southward movement of cold air from the north and west and strengthen the downdraft (Wu et al., 2017; X. Zhang et al., 2019).

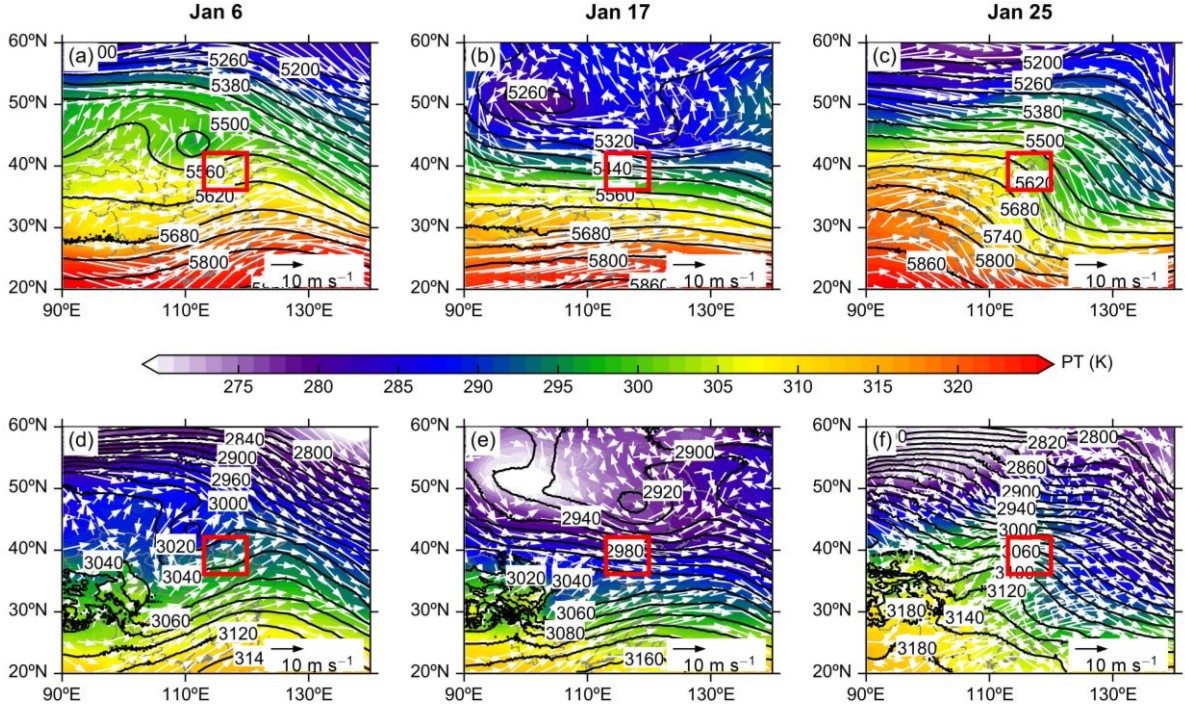

**Figure 4.** Distribution of GH (black lines), PT (shadings), and wind vectors (white arrows) at (a–c) 500 and (d–f) 700 hPa on January 6, 17, and 25. Red rectangles indicate the BTH region.

Figure 5 displays the distribution of simulated GH, PT, and wind vectors at 850 hPa level on the three days. On January 6, most of the BTH region was between two weak high pressures. Influenced by the southwesterly and southeasterly winds, the warm and humid air masses from the south and the sea were brought to the central and southern BTH. The northern BTH was mainly controlled by low pressure to the northeast, and the northwesterly winds in the area were reduced due to the blockage of the Taihang and Yanshan Mountains. On January 17, the BTH region was between the low pressure in the northeast and the high pressure in the southwest, leaving most of the BTH under the control of northwesterly winds. On January 25, eastern China was under the control of subtropical high pressure. The BTH region was located north of the subtropical high center, with southwesterly winds prevailing. Such a wind field is conducive to bringing warm and humid air masses from the south to the BTH.

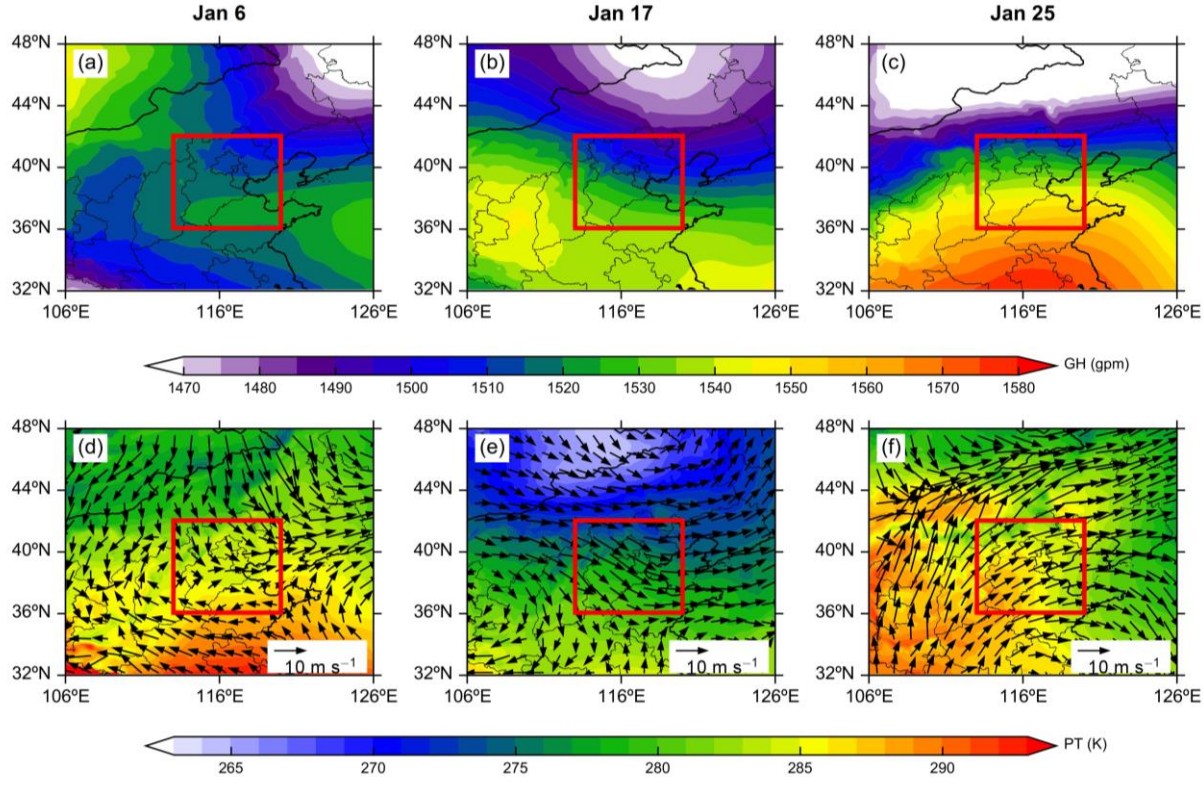


**Figure 5.** Distribution of simulated (CTL) GH (a–c), PT (d–f: shadings), and wind vectors (d–f: black arrows) at 850 hPa on January 6, 17, and 25. Red rectangles indicate the BTH region.

The northerly or northwesterly airflow in the lower troposphere tends to form a sink motion on the leeward slope after being
blocked by the Yanshan and Taihang Mountains, which will lead to a decrease in PBLH and wind speed in the plains of the
BTH region (Fig. 6). On the three days, the daily mean PBLH was generally below 300 m in the central and southern plains
of BTH, and its low-value area corresponded well to the high-value area of near-surface $PM_{2.5}$ concentrations. Moreover,
due to the disturbance of local circulation caused by the Yanshan and Taihang Mountains, the near-surface wind field
showed different distribution characteristics from the lower troposphere with wind speeds below 2 m s$^{-1}$ in most areas. On
January 6, high $PM_{2.5}$ concentrations were concentrated in eastern BTH, and a northeast-southwest transport channel was
formed under the influence of northeasterly winds. This distribution of wind field was consistent with the average result
during the three pollution periods (Fig. S3b). On January 17, northwesterly winds prevailed in the northern Beijing area due
to strong airflow in the lower troposphere. However, the blockage of the mountains made the airflow sharply weakened in
the plain area after crossing the mountains, and formed subsidence and weak divergence, which led to a large accumulation
of pollutants here. A similar transport channel was found on January 25, while its wind field was completely different from
that on January 6. Most of the plains are controlled by southerly winds. The southerly winds formed a convergence zone

with the northerly winds south of Beijing, which was not conducive to the outward dispersion of pollutants. Furthermore, the southerly winds brought warm and humid air mass from the south, which facilitated the formation of secondary pollution.

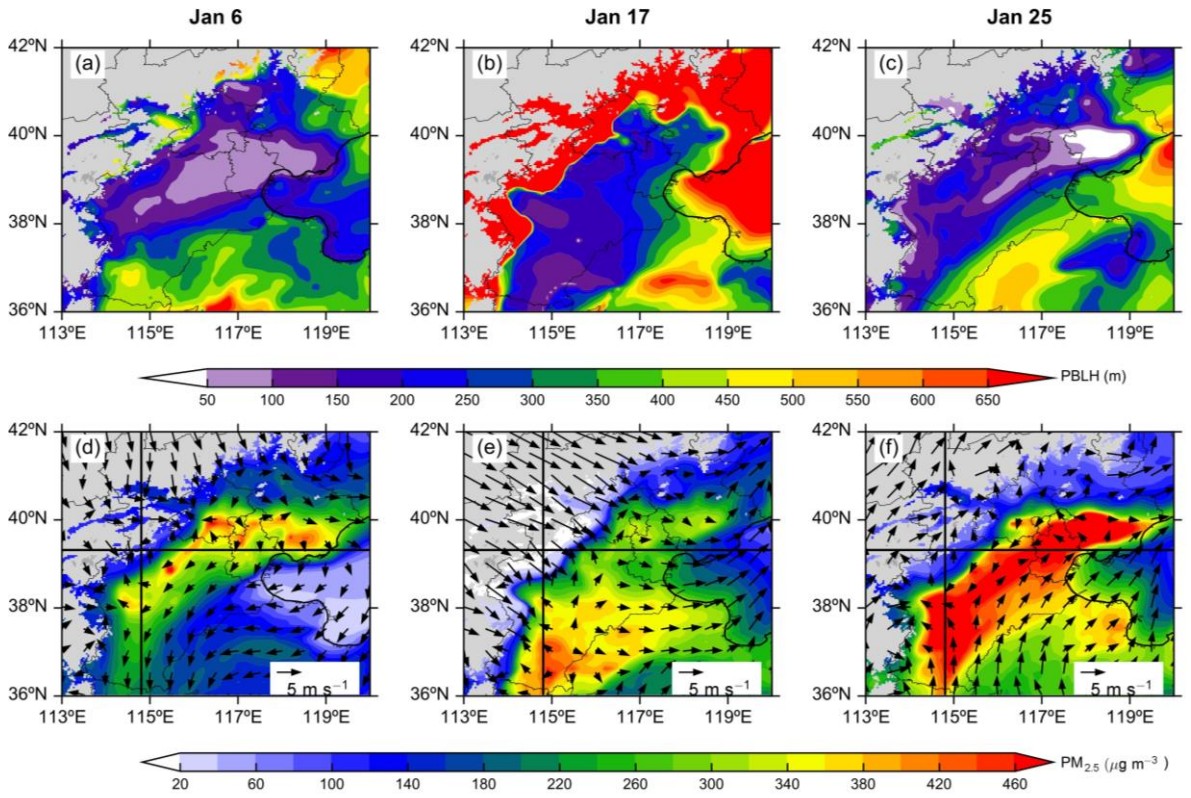

**Figure 6.** Distribution of simulated (CTL) daytime (09:00–16:00 BJT) PBLH (a–c), near-surface PM$_{2.5}$ concentrations (d–f: shadings), and wind vectors at 10 m (d–f: black arrows) on January 6, 17, and 25. The grey shadings denote the terrain height over 1000 m. The black lines indicate the location of the vertical cross-sections shown in Figures 7, 9–12.

### 3.3 Influence of local circulations on pollutant distribution

Under these weak weather-scale systems, local circulation may dominate the distribution of pollutants and the development of haze. Figure 7 displays the daytime vertical circulation vectors and PM$_{2.5}$ concentrations along the west-east (to the east of the Taihang Mountains) cross-section and along the south-north (to the south of the border between the Taihang and Yanshan Mountains) cross-section respectively on the three days. For the cross-section along the west-east, a clockwise vertical circulation was formed in the lower level on January 6. The westerly airflow sharply weakened after crossing the mountains, and the zonal wind speeds at lower levels (< 500 m, where the pollution was most severe) were mostly below 2 m s$^{-1}$; the differential heating between the mountain slopes and the plains caused the atmosphere on the slopes to rise with relative heating and the atmosphere on the plains to sink with relative cooling, resulting in a weak clockwise local circulation

between the eastern Taihang Mountains and the BTH plain ($119\,^\circ$E) (Fig. 7a); pollutants then accumulated in the PBL through this recirculation. There was a similar vertical circulation on January 25 (Fig. 7c). However, this circulation was limited (between the eastern Taihang Mountains and $118\,^\circ$E) due to the control of lower and near-surface southwesterly winds. On January 17, although a sinking motion occurred within the PBL, the zonal wind speeds were larger throughout the layer compared to the other two days, mostly ranging between 2 to 5 m s$^{-1}$ (Fig. 7b). The stronger westerly winds made it relatively easy for pollutants to disperse eastward, thus the PM$_{2.5}$ concentrations were lower than those on the other two days. For the cross-section along the south-north, the PM$_{2.5}$ concentrations on January 6 were significantly lower than those on January 17 and 25. On January 6, northeasterly winds prevailed near the surface of the center and southern BTH, and the pollutants were transported from northeast to southwest via this channel (Fig. 6d); the airflow over the mountains formed a whole layer of subsidence near $38\,^\circ$N (Fig. 7d), which inhibited the upward transport of pollutants; at the same time, there was a vertical local circulation at $33$–$37\,^\circ$N, between 700 and 1500 m (Fig. 7d), which made pollutants recirculate in this region and not easily disperse to the outside. However, due to the high altitude of this circulation, its restrictions on pollutants were not as strong as the zonal circulations on January 6 (Fig. 7a) and 25 (Fig. 7c). On January 17, a wind convergence zone accompanied by sinking motion existed in the lower levels ($<$ 1000 m) near $37\,^\circ$N (Fig. 7e). The combined effect of southerly and northerly winds made the pollutants difficult to disperse outward, thus accumulating locally. On January 25, southerly winds prevailed throughout the layer below 1500 m (Fig. 7f), which is a typical meteorological condition leading to severe air pollution in this region (Huang et al., 2020; X. Zhang et al., 2019; Zhong et al., 2018b). The southerly winds facilitated local pollutants accumulation by weakening their horizontal diffusion and bringing in warm and humid air mass.

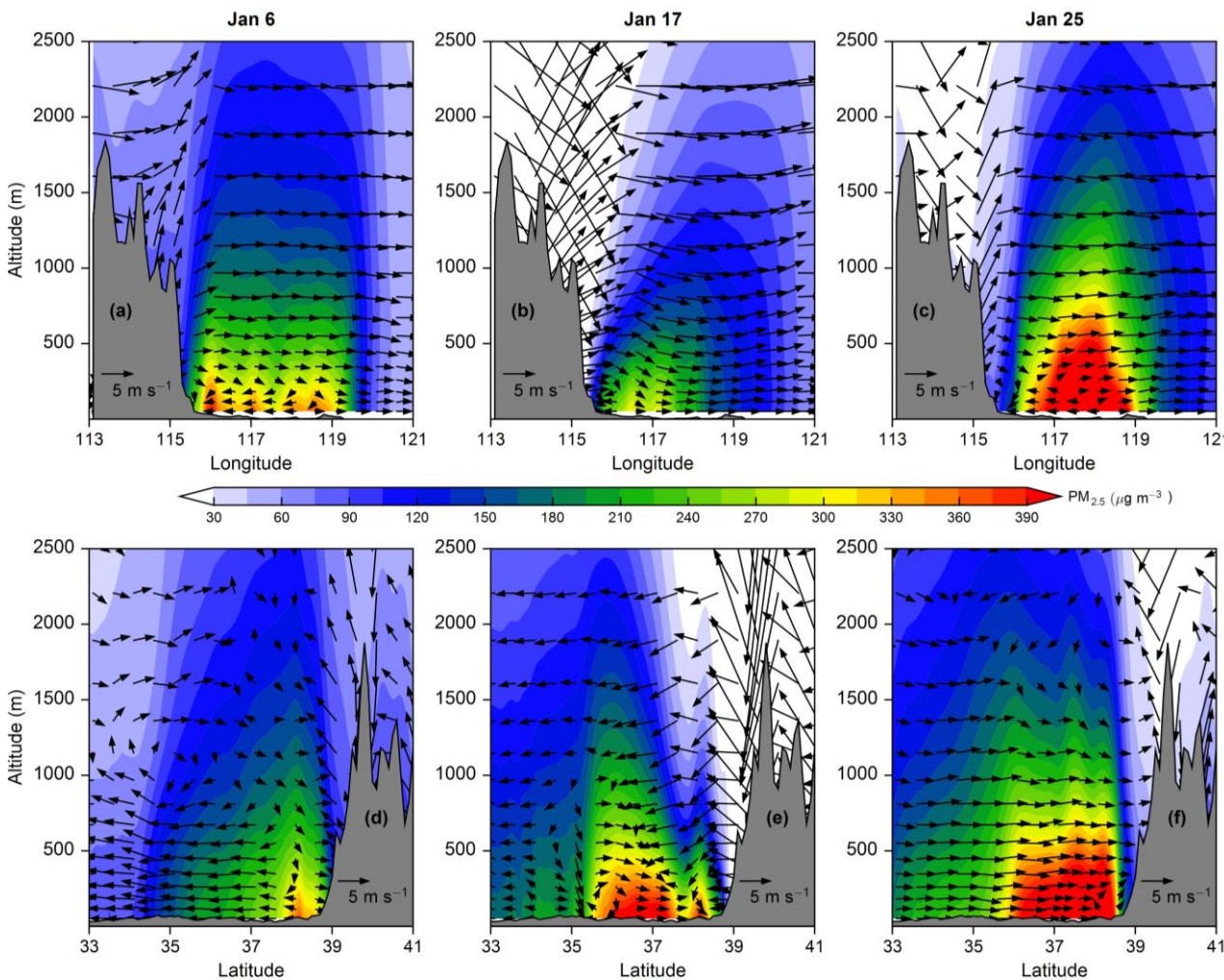

**Figure 7.** Vertical cross-section of simulated (CTL) wind field (a–c: zonal wind and 100 times of vertical velocity; d–f: meridional wind and 100 times of vertical velocity) and PM$_{2.5}$ concentrations during daytime (09:00–16:00 BJT) on January 6, 17, and 25.

**3.4 Amplification of local circulations on heavy haze by ARI**

Given the considerable impacts of local circulation and ARI on the distribution of PM$_{2.5}$ extremes, we further analyzed the potential link between them. According to the location of the cross-sections in Fig. S3, Fig. 8 shows the vertical distribution of wind field and PM$_{2.5}$ with and without the ARI mechanism and the difference of horizontal wind speeds induced by ARI. For the cross-section along the west-east, ARI strengthened the clockwise vertical circulation near 500 m by simultaneously

enhancing the westerly winds in the upper level (500~1000 m) and the easterly winds in the lower level (< 500 m) (Fig. 8a–

c). For the cross-section along the south-north, ARI strengthened the circulation at high altitudes between 33 and 35 °N by enhancing the upper (900~1500 m) southerly winds and the lower (< 500 m) northerly winds (Fig. 8d–f). Moreover, the southerly winds in the circulation formed a wind convergence zone with the northerly winds on its north side (Fig. 8d–e), and ARI strengthened this zone by increasing the wind on both sides (Fig. 8f).

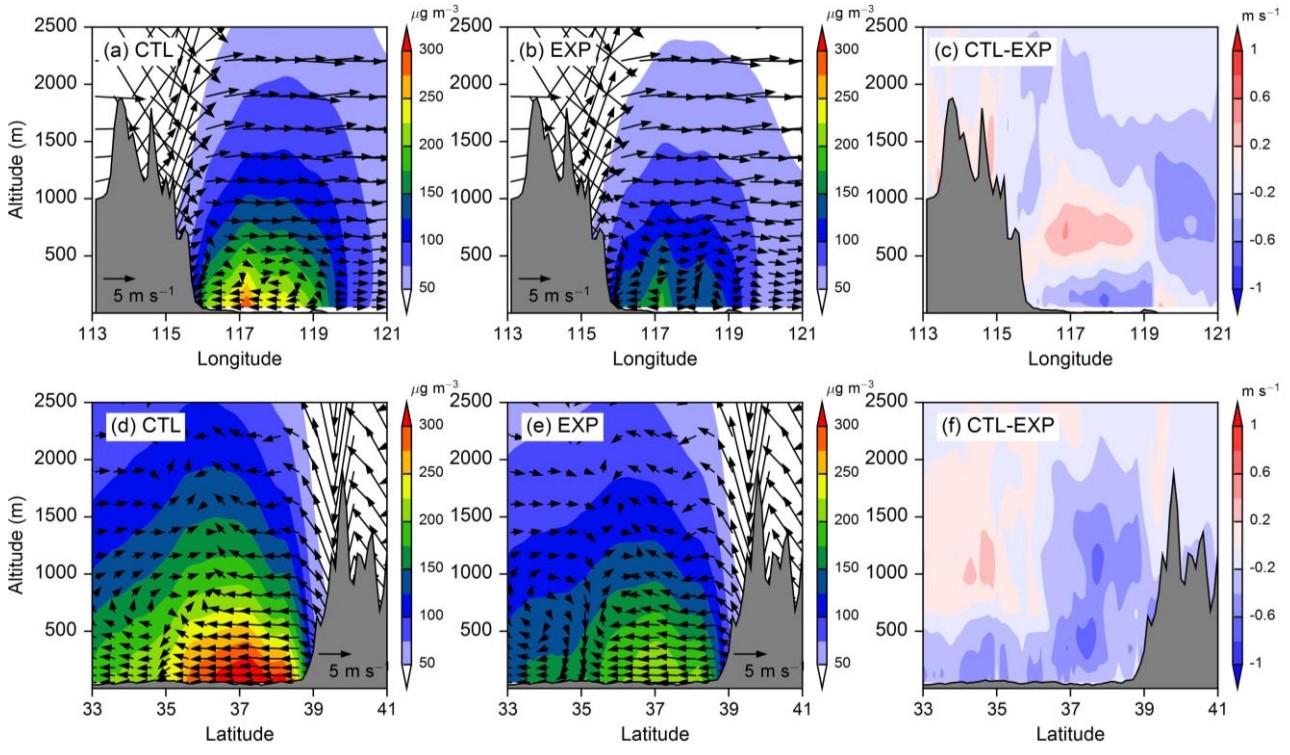

**Figure 8.** Vertical cross-section of simulated (CTL and EXP) daytime (09:00–16:00 BJT) wind field (a–b: zonal wind and 100 times of vertical velocity; d–e: meridional wind and 100 times of vertical velocity) and PM$_{2.5}$ concentrations, and the differences (CTL-EXP) of horizontal wind (c: zonal wind; f: meridional wind) induced by ARI during the three pollution periods.

As the local circulation characteristics were different on the three heavy haze days (Fig. 7), it was necessary to investigate the impact of ARI on each day. As shown in Figs. 9–12, the impacts of ARI on circulations could be broadly classified into two types: strengthened local circulation and weakened horizontal diffusion. First, ARI significantly strengthened the vertical zonal circulations on January 6 and 25: the high aerosol concentrations concentrated on the BTH plain during the daytime substantially cooled the lower atmosphere by absorbing and scattering solar radiation, and the widening difference in atmospheric heating between the mountain slopes and the BTH plain led to a simultaneous strengthening of westerly winds in the upper level and easterly winds in the lower level. On January 6, between the eastern Taihang Mountains and 119 °E, ARI increased the westerly winds (300~800 m) and the easterly winds (< 300 m) by 0.8 and 0.3 m s$^{-1}$, respectively,

on average (Figs. 9a, 11a, 11d). On January 25, ARI increased the westerly winds by 0.6 m s$^{-1}$ at the same location as on January 6; meanwhile, the airflow below 300 m changed from weak westerly to easterly under the ARI effect, forming a closed circulation with the upper westerly winds (Figs. 9c, 11c, 11f). Moreover, ARI could change the altitude of circulation. On January 6, ARI shifted the vertical circulation downward by about 100 m according to the wind speed minimum and the height of the wind shear (Fig. 11a, d). The stronger and lower vertical circulation caused further accumulation of pollutants in the lower level, which then led to substantial cooling of the lower atmosphere (Fig. 9d, f) and weak vertical turbulent diffusion (Fig. 9g, i). ARI also strengthened the meridional circulations on January 6, although it did not cause the same downshift as the zonal circulations. Due to the ARI effect, the southerly winds in the upper level and the northerly winds in the lower level were strengthened simultaneously (Figs. 10a, 12a, 12d); the strengthened vertical circulation in this local area likewise traps the pollutants in the limited space, which further cooled atmosphere (Fig. 10d) and weakened turbulent diffusion (Fig. 10g) in lower atmosphere.

Second, ARI weakened the horizontal diffusion of pollutants. For the relatively lightly polluted northern BTH on January 17, ARI weakened the westerly winds below 300 m and east of 117 ° E (Figs. 9b, 11b, 11e), with a maximum wind speed reduction of 1 m s$^{-1}$. In addition, ARI enhanced the sinking of airflow near 116 ° E (Fig. 11b, e), promoting the accumulation of aerosols in the lower layer. For the heavily polluted southern BTH on January 17, ARI enhanced the wind convergence zone near 37 ° N by simultaneously strengthening the southerly winds south of the convergence line and the northerly winds north of the convergence line (Figs. 10b, 12b, 12e). At the same time, ARI pushed this convergence line northward, causing southerly winds to prevail below 200 m over the plain. In addition, ARI enhanced the southerly winds below 600 m along the south-north on January 25 (Figs. 10c, 12c, 12f), with an average increase of 0.6 m s$^{-1}$. Pollutants accumulated locally due to the blockage of the Yanshan Mountains, which then led to the cooling of the lower atmosphere (Fig. 10e–f) and weakening of the vertical turbulent motion (Fig. 10h–i).

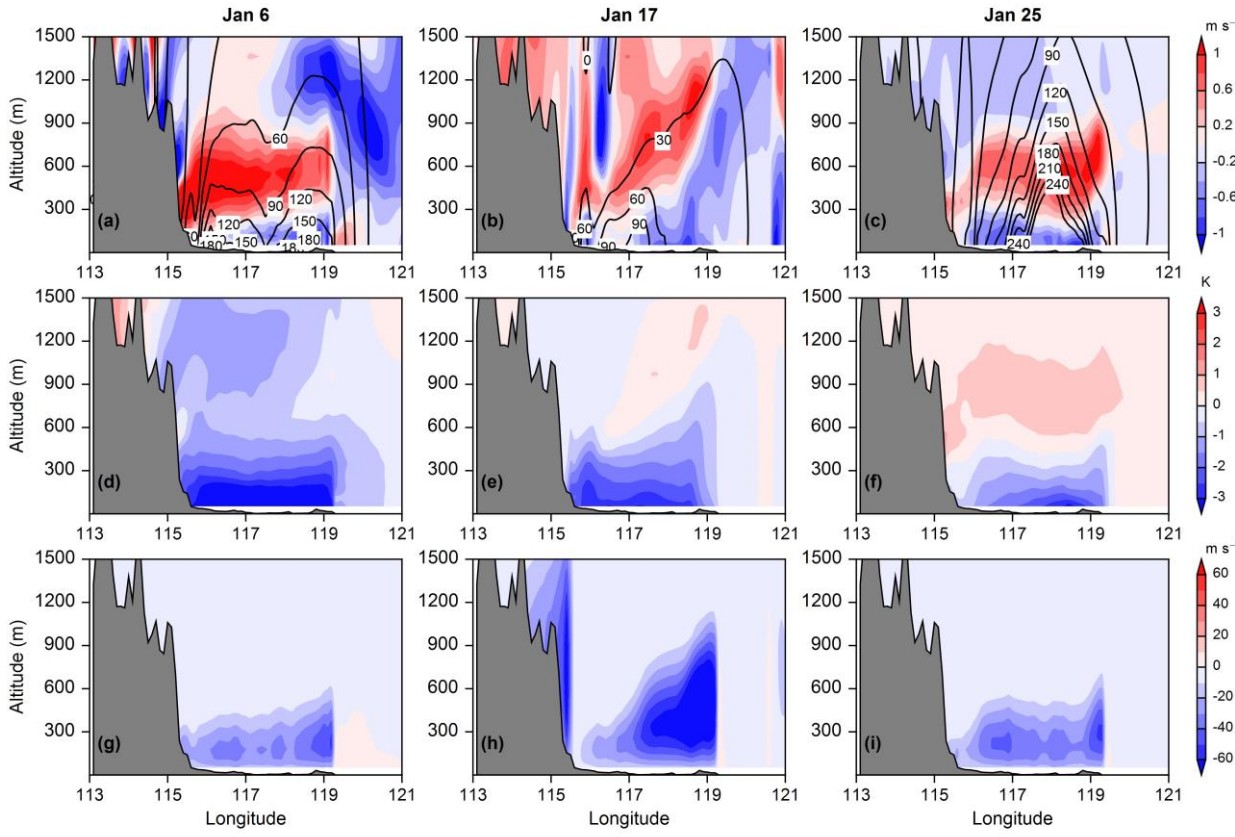

**Figure 9.** Vertical cross-section of simulated (a–c) zonal wind (shadings) and PM$_{2.5}$ concentration (contours: μg m$^{-3}$), (d–f) PT, and (g–i) vertical turbulent diffusion coefficient differences (CTL–EXP) induced by ARI during daytime (09:00–16:00 BJT) on January 6, 17, and 25.

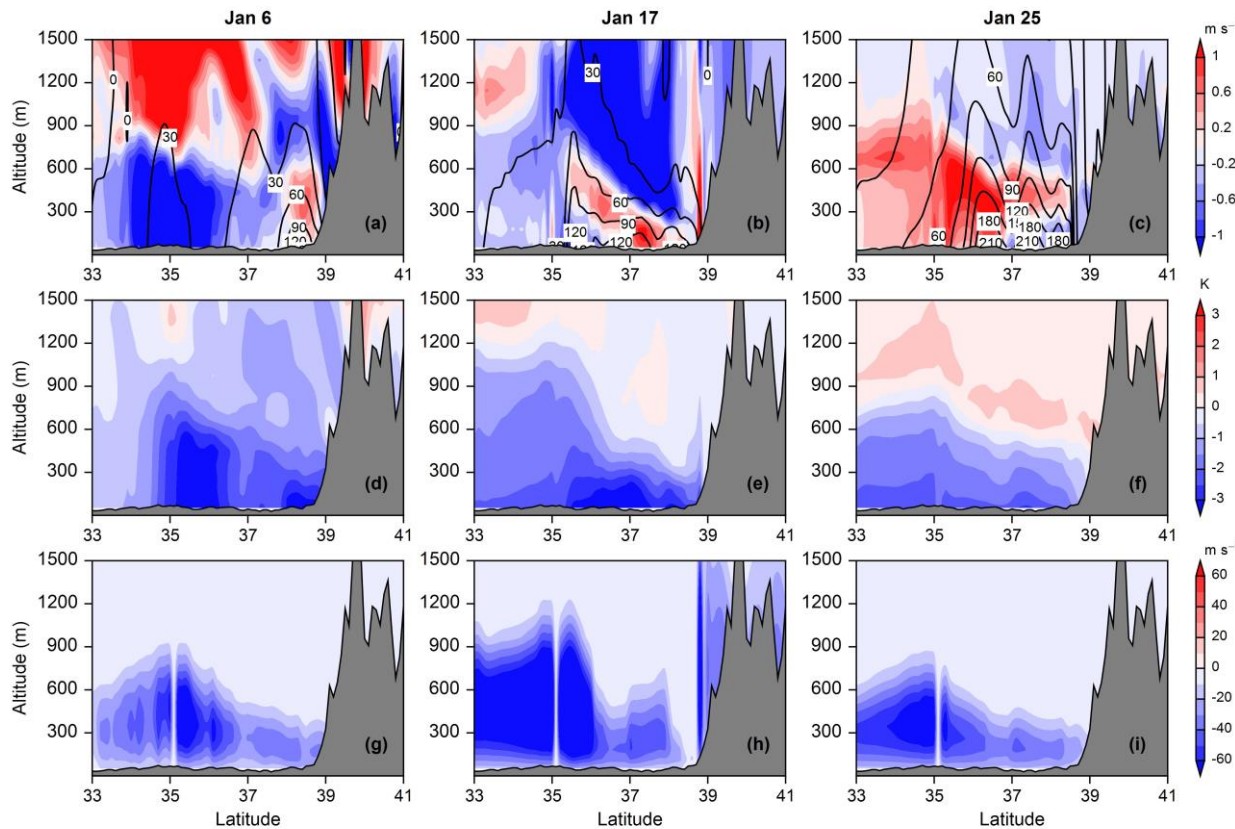

**Figure 10.** Vertical cross-section of simulated (a–c) meridional wind (shadings), PM$_{2.5}$ concentration (contours: µg m$^{-3}$), (d–f) PT, and (g–i) vertical turbulent diffusion coefficient differences (CTL–EXP) induced by ARI during daytime (09:00–16:00 BJT) on January 6, 17, and 25.

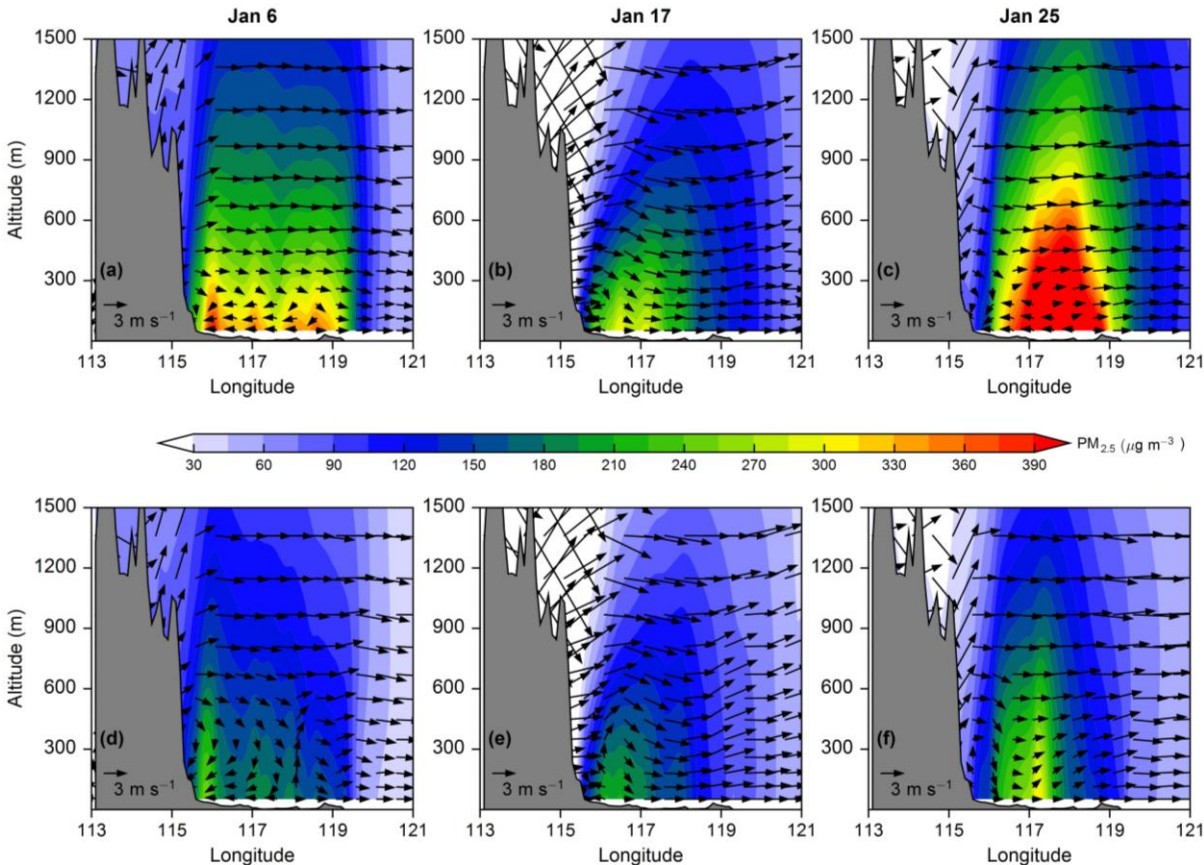

**Figure 11.** Vertical cross-section of simulated wind field (zonal wind and 100 times of vertical velocity) and PM$_{2.5}$ concentrations from (a–c) CTL and (d–f) EXP during daytime (09:00–16:00 BJT) on January 6, 17, and 25.

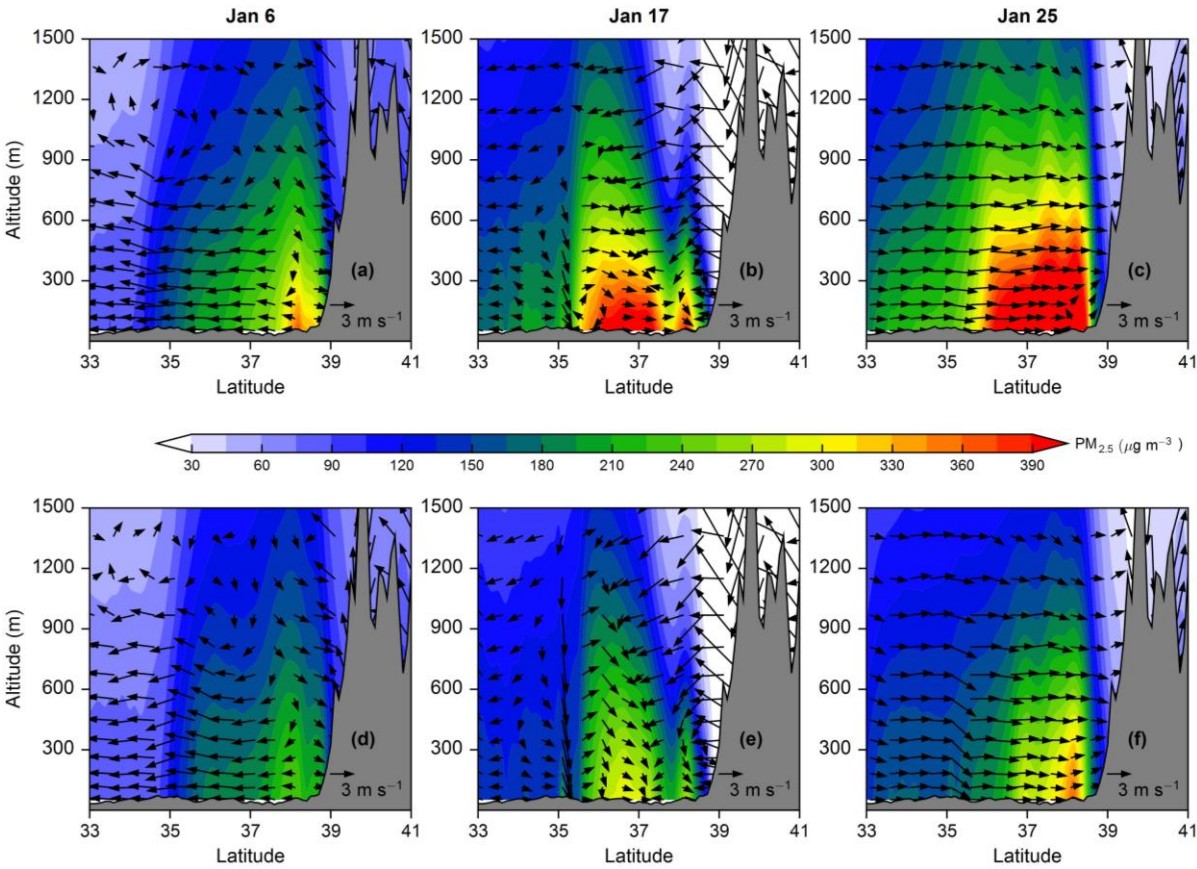

**Figure 12.** Vertical cross-section of simulated wind field (meridional wind and 100 times of vertical velocity) and PM$_{2.5}$ concentrations from (a–c) CTL and (d–f) EXP during daytime (09:00–16:00 BJT) on January 6, 17, and 25.

## 4 Conclusions

In this study, the link between aerosol, local vertical circulation, and heavy haze pollution in the BTH plain in winter was investigated, based on surface and sounding observations and simulation experiments by the atmospheric chemistry model GRAPES_Meso5.1/CUACE in January 2017.

    From a synoptic perspective, the appropriate configurations of the upper, middle, and lower levels provided favorable conditions for the accumulation of pollutants. During the haze pollution, the BTH region was mainly controlled by the zonal

westerly airflow or northwesterly airflow in the upper and middle troposphere, and the most polluted central and southern BTH was often dominated by the southwesterly winds in the lower troposphere; at the same time, the blockage of the Taihang and Yanshan Mountains significantly weakened airflow from the west and north, while hindering the northward and westward transport of pollutants.

Under these unfavorable synoptic conditions, the typical local circulation induced by the mountainous topography played a key role in the heavy haze pollution. During the daytime on haze days, a local closed vertical circulation or wind convergence zone was formed in the lower atmosphere between the mountain slopes and the BTH plain under the influence of the mountainous topography, which was not conducive to the pollutant vertical and horizontal diffusion. Both the intensity and location of the vertical circulation played an important role in the pollutant distribution. The circulation with high intensity and low altitude could constrain near-surface pollutants to a more limited area. More importantly, the superposition of the ARI mechanism and local circulation could significantly aggravate haze pollution. According to the simulation results of this study, ARI mainly amplified the impacts of local vertical circulation on haze in two ways: strengthening local circulation and weakening horizontal diffusion. For the clockwise vertical circulation, ARI not only strengthened both the upper westerly winds and the lower easterly winds, but also pressed the circulation toward the lower atmosphere; for the wind convergence zone, ARI strengthened the southerly and northerly winds on both sides of the convergence line and simultaneously pushed the convergence line move northward. ARI amplified the inhibitory of local circulation on vertical and horizontal diffusion of pollutants through these two pathways, leading to pollutants recirculating in a more limited space. With the superposition of ARI and local circulation, aerosols accumulated rapidly in the lower atmosphere, which led to more stable atmospheric stratification and subsequent deterioration of haze pollution.

**Data availability**

All raw data can be provided by the corresponding authors upon request.

**Author contribution**

HW and XZ conceived the idea; YP and HW designed the experiment; YP ran the model and wrote the manuscript draft; ZL and WZ provided the observation data and helped perform the analysis with constructive discussions; YP, SL, and CH analysed the data; YP, HW, and HC reviewed and edited the manuscript.

**Competing interests**

The authors declare that they have no conflict of interest.

**Financial support**

This work was supported by the NSFC Major Project (42090030), the National Key Research and Development Program of China (2019YFC0214601), and the NSFC for Distinguished Young Scholars (41825011).

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
