# Peer review of "Superimposed effects of typical local circulations driven by mountainous topography and aerosol-radiation interaction on heavy haze in the Beijing-Tianjin-Hebei central and southern plains in winter"

_EGUsphere, 2022_

## Author Comment (AC1)

**Authors' Response to Referee #2**

**General comments:**

The authors of the manuscript entitled: "Superimposed effects of typical local circulations driven by mountainous topography and aerosol-radiation interaction on heavy haze in the Beijing-Tianjin-Hebei central and southern plains in winter", try to investigate the link between aerosol, local vertical circulation, and heavy haze pollution in the Beijing-Tianjin-Hebei plain in winter, by implementing the atmospheric chemistry model GRAPES_Meso5.1/CUACE in January 2017. In my opinion this is an interesting manuscript suitable for Atmospheric Chemistry and Physics journal, however some important issues need to be addressed before it can be further considered for possible publication.

**AC:** We sincerely thank the referee for taking time to carefully read through the manuscript, recognition of our work, and insightful comments and suggestions, which greatly improved the substance of the study. Our manuscript is revised according to all the comments from the referee. All changes are highlighted in red in the revised manuscript. Our point-by-point responses to all the comments are provided below. We sincerely hope the corrections will meet with approval.

**Specific comments:**

**RC1:** Lines 20-21: This phrase is not clear. Please revise.

**AC1:** We corrected this phrase (Lines 20–22) as well as similar expressions throughout the text in the revised manuscript.

**RC2:** Line 33: Please define $PM_{2.5}$ based on their diameter.

**AC2:** We rewrote this sentence based on the diameter of $PM_{2.5}$ in Lines 34–35 in the revised manuscript.

**RC3:** Line 43: The expression "high concentration aerosols" is not clear. Please revise.

**AC3:** This sentence was corrected in revised manuscript (Lines 47–48).

**RC4:** Line 89: Why was this time period selected? This must be explained.

**AC4:** We appreciate the referee's comment. During the period from December 2016 to February 2017, the Beijing-Tianjin-Hebei (BTH) region suffered from several heavy pollution episodes, of which January 2017 was the most representative month with persistent and heavy pollution levels. Therefore, we selected this month to investigate the link between aerosol, local vertical circulation, and heavy haze pollution. We have added this explanation in Lines 74–75 in the revised manuscript.

**RC5:** Figure 1: The abbreviation OBS is not defined in the legend. Please revise.

**AC5:** The description of OBS has been added in the caption in Figure 1 in the revised manuscript.

**RC6:** Line 127: Please define the type of these correlation coefficients.

**AC6:** We added the description of the correlation coefficient (r) in Section 2.3 in the revised manuscript.

**RC7:** Lines 128-129: The difference of the correlation coefficients (0.03) between the two numerical scenarios (CTL and EXP) is really low to support this statement. Please revise.

**AC7:** Yes. The difference of the r values between CTL and EXP is very small, and this metric alone does not fully indicate that the simulations agree with observations. Therefore, we introduced multiple metrics including the mean bias (MB), the root mean square error (RMSE), the mean fractional bias (MFB), and the mean fractional error (MFE) besides r to evaluate the model performance (Lines 117–119). The statistical results showed that CTL with ARI is closer to the observations compared to EXP, with higher r (from 0.71 to 0.74), smaller MB (from -40.2 to -16.4 μg m$^{-3}$), and smaller RMSE (from 57.0 to 45.3 μg m$^{-3}$). Besides, both MFB and MFE showed substantial reductions, from -34.2 μg m$^{-3}$ and 37.6 μg m$^{-3}$ to -15.7 μg m$^{-3}$ and 28.5 μg m$^{-3}$, respectively. According to model performance goals for PM$_{2.5}$ proposed by Morris (2005), both CTL and EXP achieved an average level (MFB ≤ ±60% and MFE ≤ 75%), while CTL exceeded a good level (MFB ≤ ±30% and MFE ≤ 50%) and was very closed to an excellent level (MFB ≤ ±15% and MFE ≤35%). We have revised the original phrases according to the above contents in the revised manuscript (Lines 157–164) and added Table 1 to make the comparison between CTL and EXP more intuitive.

**RC8:** Lines 131-134: The justification for the selection of the three studied pollution periods (January 5–7, 16–18, and 23–26) is not adequate. Other extreme pollution periods are also indicated in Figure 2. In addition, the selection of January 6, 17, and 24 as the representatives of the three pollution periods is also not adequately justified. This is an important issue because biased results can be implied. Please include more convincing explanations. Moreover, in my opinion the authors should expand the implementation of their method in other time periods besides January 2017 to enhance their manuscript.

**AC8:** We have extended the first pollution period from January 5–7 to January 1–7, covering the days with high PM$_{2.5}$ concentrations at the beginning of the month. In addition, more convincing explanations were introduced for the selection of the three days. These revisions can be found in Lines 165–178 in the revised manuscript. It should be noted that according to the definition of heavy pollution (the daily mean PM$_{25}$ concentration larger than 150 μg m$^{-3}$) issued by the Ministry of Environmental Protection of China, we replaced the analysis on January 24 with the results on January 25 in the full text. But the good news is the strengthening of the local circulation by ARI is more pronounced due to the higher PM$_{2.5}$ concentrations on the 25$^{th}$ than on the 24$^{th}$. This result again supported the findings of this study.

To expand the implementation of our method and make the conclusion more representative, we added the average condition of all three pollution periods (January 1–7, 16–18, and 23–26) in January 2017. The average result was highly consistent with the findings of the three days (Lines 311–325), which shows that our analysis is representative in this region. Due to the limitation of space and structure, we did not add other time periods besides January 2017 to this manuscript. However, we still thank the referee's valuable suggestions, and we will continue to explore this issue in the future work, and carry out statistics and comparative analysis of the interaction between aerosols and local circulation in

different seasons and years.

**RC9:** Figure 4: The quality of these images is very low and has to be improved. In addition, some elements are not defined the legend. More specifically: a) which images correspond at 500 hPa and which at 700 hPa? b) the red square indicates the BTH region?

**AC9:** We have improved the resolution of all figures including Figure 4. Moreover, the description of Figure 4 has been supplemented in the revised manuscript according to the referee's comments.

**RC10:** Figure 5: A red square is also needed here to indicate the BTH region. Please add it.

**AC10:** The red square has been added to Figure 5 in the revised manuscript.

**RC11:** Lines 231-232: This sentence is incomplete

**AC11:** This sentence was corrected in the revised manuscript (Lines 311–312).

**RC12:** Lines 238-242: This section is not clear especially for January 24. Please revise.

**AC12:** We have modified this section in the revised manuscript (Lines 334–340).

**RC13:** Line 248: The enhancement of northerly winds in the lower level on January 24 is not clear in Figure 9c. Please revise.

**AC13:** This section was revised in Line 346 in the revised manuscript as we replaced the original analysis of January 24 with January 25.

**RC14:** Line 253: A reference to Figure 10 (b, e) would be useful here

**AC14:** We added this in the revised manuscript (Line 351).

**RC15:** Figure 9: (g–i) is missing from the legend. Please include it.

**AC15:** (g–i) was added in the Figure 10 in the revised manuscript (Line 369).

**References:**

Morris, R. E., McNally, D. E., Tesche, T. W., Tonnesen, G., Boylan, J. W., and Brewer, P.: Preliminary Evaluation of the Community Multiscale Air Quality Model for 2002 over the Southeastern United States, J. Air Waste Manage. Assoc., 55(11), 1694–1708, https://doi.org/10.1080/10473289.2005.10464765, 2005.

---

## Author Comment (AC2)

**Authors' Response to Referee #1**

**RC1:** The current paper is focused on the air pollution by $PM_{2.5}$ at the Beijing-Tianjin-Hebei region in China. The subject merits to be investigated due to the noticeable impact on the affected population. The period investigated is around January 2017, although only three days are selected. Measurements are considered together with modelling analysis. Observations are provided by the National Environmental Monitoring Center, 149 stations, and the Hebei Meteorological service, 210 stations. Moreover, two kind of modelling calculations are used, one of them with the aerosol-radiation interaction, and the second calculation without this interaction. The synoptic pattern is presented at varied pressure surfaces, and vertical cross-sections with the airflow and concentration are also shown. Although the subject and procedure are suitable for a publication, some restrictions of this research indicate that this paper could be accepted in a journal with low impact, but not in Atmospheric Chemistry and Physics.

**AC1:** We sincerely thank Pérez for taking time to carefully read through the manuscript, constructive comments and suggestions, which greatly improved the substance of our study. Our manuscript is revised according to all your valuable comments, and the changes are highlighted in red in the revised manuscript. To achieve the publication standards of Atmospheric Chemistry and Physics, we made Major Revision to this manuscript (summarized one-by-one as follows):

(1) We revised the description of the importance and necessity for this study in the introduction and added relevant references (Lines 42–43, 53–54, 69–73).

(2) The analysis of the average results for the three pollution periods was added in the revised manuscript (Lines 190–191, 208–219, 311–325) and the Supplement (Fig. S1–S3) besides the three haze days, and extended the first pollution period from January 5–7 to January 1–7 (Lines 167–172).

(3) The analysis for January 25 with higher $PM_{2.5}$ concentrations was introduced to replace the results for January 24 (Lines 224–225, 240–242, 260–264, 284–286, 298–302, 335–338, 355–357).

(4) The information of emission was added (Lines 37–41, 105–106, 116).

(5) Some statistical parameters, Table 1, and related instructions for $PM_{2.5}$ evaluation were added (Lines 117–119, 157–164, 184).

Other minor revisions:

(1) The names of mountain ranges and sea were introduced in Fig. 1a.

(2) The format and spelling errors throughout the full text including references were checked and corrected.

(3) All the figures involving elements of January 24 have been replaced with those of January 25.

(4) The conclusion and the abstract have been revised according to the changes in the text.

Aerosol-radiation interaction (ARI) not only play an important role in the regulation of the global climate system, but may also lead to complex feedbacks on regional or local scales (Ramanathan et al., 2001; Shindell and Faluvegi, 2009). In recent years, researchers achieved many results around ARI in revealing the formation of heavy air pollution in China, mainly including the interactions between ARI, the planetary boundary layer (PBL), and long-range transport (Gao et al., 2015; Ding et al., 2016; Li et al., 2017; Wang et al., 2015, 2018; Huang et al., 2018, 2020), and these works have been published in Atmospheric Chemistry and Physics and other high-impact journals. However, these works rarely focus on the link between ARI and local circulation driven by typical topography. For the Beijing-Tianjin-Hebei (BTH) region, the mountainous topography is an important factor contributing to its persistent severe air pollution (Chen et al., 2009; Liu et al., 2009; Miao et al., 2015; Zhang et al., 2018). Based on the above studies, this paper further discussed the interaction between ARI and local circulation driven by mountainous topography, and found that their superimposed effects may be the most important reason for the extreme haze events in critical polluted region. We believe that this study of ARI on local circulation is an effective complement to the ARI studies on multiple scales and it is as important as the results from global to regional scales mentioned above. This research contributes to a more comprehensive and accurate understanding of the causes of heavy pollution.

**RC2:** The main inconvenience lies on the extremely low number of situations where the study is made, only three days, 6[th], 17[th], and 24[th]. Although the analysis is detailed, the readers should know if these days are representative enough for the pollution days at this site. Moreover, the readers should know if these conditions could be reproduced at different sites.

**AC2:** In addition to the three pollution days, we added the average condition of all three pollution periods in January 2017 to investigate the general link between local circulation, ARI, and haze pollution. The average result was highly consistent with the findings of the three days (Lines 311–325 in the revised manuscript), which shows that our analysis is representative in this region. Moreover, to make this paper structure more logical, we also added descriptions of the average weather situation for the three pollution periods in the text (Lines 208–219) and the Supplement (Fig. S1-S3). Nevertheless, average result failed to reflect more characteristics about the local circulation under different haze days, such as the widespread westerly winds along the west-east cross-section on January 17 and the widespread southerly winds along the north-south cross-section on January 25, suggesting that the long-term average result may weaken the features on local scale. Therefore, a detailed analysis of individual day in indispensable. It should be noted that according to the definition of heavy pollution (the daily mean $PM_{25}$ concentration larger than $150 \ \mu g \ m^{-3}$) issued by the Ministry of Environmental Protection of China, we replaced the analysis on January 24 with the results on January 25 in the full text. But the good news is the strengthening of the local circulation by ARI is more pronounced due to the higher $PM_{2.5}$ concentrations on the 25[th] than on the 24[th]. This result again proves that our findings are representative.

We understand that readers should know more conditions at different sites. However, the observed

vertical data is very scarce and there are only three weather sounding stations (Beijing, Tangshan, and Xingtai) in our study area. Moreover, we focused on the central and southern plains of Beijing-Tianjin-Hebei associated with the local circulation and heavy pollution, rather than on specific sites. The evaluation of the vertical potential temperature at the only three sounding sites showed a well reproduction of atmospheric vertical structure by the model, which provided a reasonable prerequisite for the subsequent local circulation analysis.

**RC3:** Since the pollution levels are affected by factors such as the emissions and the meteorological variables, some information about the patterns of both factors could be useful to focus the pollution problem at the site.

**AC3:** Yes. Anthropogenic emissions and meteorological conditions are two key factors affecting pollutants. We have added some description of the impact of emissions on the heavy pollution (Lines 37–41) and information on the emission inventory used in the model (Lines 105–106, 116) in the revised manuscript. Numerous previous studies have shown that heavy winter pollution in northern China is often caused by a combination of high emissions and unfavorable meteorological conditions. However, for short period, emissions in a region do not normally change much, and the regional or local meteorological conditions may dominate pollution levels (Wang et al., 2018; Zheng et al., 2015; Zhong et al., 2017). Moreover, pollutant emissions are quite complex and obtaining accurate hour-by-hour pollutant emission data is very difficult. The emission inventory used in our model is the monthly average result and we used it for both sets of experiments.

**RC4:** Figure 2 presents the concentration evolution. The authors should comment the reason to discard the first days of the month when the concentrations are even higher than those selected for the analysis.

**RC4:** We have extended the first pollution period from January 5–7 to January 1–7, covering the days with high $PM_{2.5}$ concentrations at the beginning of the month. The reason to discard the first days of the month was that we did not consider it to be a complete continuous rise in pollution. However, this reason was not sufficient since the pollution levels on the first four days (January 1–4) were comparable to those on the last three days (January 5–7) of the period, so we revised it based on the referee's comments. The revisions can be seen in Lines 167–172 in the revised manuscript.

**RC5:** Some statistics to contrast the measured and calculated concentrations should be introduced. If the correlation is made with the Pearson correlation coefficient, the authors should consider that a good value of this estimator could not indicate a good agreement between measured and calculated values. A better statistic for this calculation could be the index of agreement.

**AC5:** We thank the referee's suggestion. More statistical parameters including the mean bias (MB), the mean fractional bias (MFB), the mean fractional error (MFE), and the root mean square error (RMSE) were introduced (Lines 117–119) to make the evaluation on $PM_{2.5}$ concentration more comprehensive. As list in Table 1, the statistics showed that, compared to the results without ARI (the EXP case), the

model considering ARI (the CTL case) showed better agreements with the observations, with reduced MB (from -40.2 to -16.4 µg m$^{-3}$), reduced MFB (from -34.2% to -15.7%), reduced MFE (from 37.6% to 28.5%), reduced RMSE (from 57.0 to 45.3 µg m$^{-3}$) and increased r (from 0.71 to 0.74). We added Table 1 and the related description (Lines 157–164) in the revised manuscript.

Table 1. Model evaluation for PM$_{2.5}$ in BTH during January 2017.

|  | r | MB µg m$^{-3}$ | RMSE µg m$^{-3}$ | MFB % | MFE % |
|---|---|---|---|---|---|
| CTL | 0.74 | -16.4 | 45.3 | -15.7 | 28.5 |
| EXP | 0.71 | -40.2 | 57.0 | -34.2 | 37.6 |

**Minor remarks**.

**RC1:** The names of mountain ranges and sea should be introduced in Fig. 1a (indicated in the text, l. 51), not in Fig. 1b.

**AC1:** We have corrected this error in Fig. 1a of the revised manuscript.

**References**

Chen, Y., Zhao, C., Zhang, Q., Deng, Z., Huang, M., and Ma, X.: Aircraft study of mountain chimney effect of Beijing, China, J. Geophys. Res., 114, D08306, https://doi.org/10.1029/2008JD010610, 2009.

Ding, A. J., Huang, X., Nie, W., Sun, J. N., Kerminen, V. M., Petäjä, T., Su, H., Cheng, Y. F., Yang, X.-Q., Wang, M. H., Chi, X. G., Wang, J. P., Virkkula, A., Guo, W. D., Yuan, J., Wang, S. Y., Zhang, R. J., Wu, Y. F., Song, Y., Zhu, T., Zilitinkevich, S., Kulmala, M., and Fu, C. B.: Enhanced haze pollution by black carbon in megacities in China, Geophys. Res. Lett., 43(6), 2873–2879, https://doi.org/10.1002/2016GL067745, 2016.

Gao, Y., Zhang, M., Liu, Z., Wang, L., Wang, P., Xia, X., Tao, M., and Zhu, L.: Modeling the feedback between aerosol and meteorological variables in the atmospheric boundary layer during a severe fog–haze event over the North China Plain, Atmos. Chem. Phys., 15, 4279–4295, https://doi.org/10.5194/acp-15-4279-2015, 2015.

Huang, X., Ding, A., Wang, Z. Ding, K., Gao, J., Chai, F., and Fu, C.: Amplified transboundary transport of haze by aerosol–boundary layer interaction in China, Nat. Geosci., 13, 428–434, https://doi.org/10.1038/s41561-020-0583-4, 2020.

Huang, X., Wang, Z., and Ding, A.: Impact of aerosol-PBL interaction on haze pollution: Multiyear observational evidences in North China, Geophys. Res. Lett., 45(16), 8596–8603, https://doi.org/10.1029/2018GL079239, 2018.

Li, Z., Guo, J., Ding, A., Liao, H., Liu, J., Sun, Y., Wang, T., Xue, H., Zhang, H., and Zhu, B.: Aerosol and boundary-layer interactions and impact on air quality, Natl. Sci. Rev., 4, 810–833, https://doi.org/10.1093/nsr/nwx117, 2017.

Liu, S., Liu, Z., Li, J., Wang, Y., Ma, Y., Sheng, L., Liu, H., Liang, F., Xin, G., and Wang, J.: Numerical simulation for the coupling effect of local atmospheric circulations over the area of Beijing, Tianjin and Hebei Province, Sci. China Ser. D-Earth Sci, 52, 382–392, https://doi.org/10.1007/s11430-009-0030-2, 2009.

Miao, Y., Liu, S., Zheng, Y., Wang, S., Chen, B., Zheng, H., and Zhao, J.: Numerical study of the effects of local atmospheric circulations on a pollution event over Beijing–Tianjin–Hebei, China, J. Environ. Sci, 30, 9–20, http://dx.doi.org/10.1016/j.jes.2014.08.025, 2015.

Ramanathan, V., Crutzen, P. J., Kiehl, J. T., and Rosenfeld, D.: Aerosols, Climate and the Hydrological Cycle, Science, 294, 2119–2124, https://doi.org/10.1126/science.1064034, 2001.

Shindell, D., and Faluvegi, G.: Climate response to regional radiative forcing during the twentieth century, Nat. Geosci., 2, 294–300, https://doi.org/10.1038/ngeo473, 2009.

Wang, H., Peng, Y., Zhang, X., Liu, H., Zhang, M., Che, H., Cheng, Y., and Zheng, Y.: Contributions to the explosive growth of $PM_{2.5}$ mass due to aerosol–radiation feedback and decrease in turbulent diffusion during a red alert heavy haze in Beijing–Tianjin–Hebei, China, Atmos. Chem. Phys., 18, 17717–17733, https://doi.org/10.5194/acp-18-17717-2018, 2018.

Wang, H., Shi, G. Y., Zhang, X. Y., Gong, S. L., Tan, S. C., Chen, B., Che, H. Z., and Li, T.: Mesoscale modelling study of the interactions between aerosols and PBL meteorology during a haze episode in China Jing–Jin–Ji and its near surrounding region – Part 2: Aerosols' radiative feedback effects, Atmos. Chem. Phys., 15, 3277–3287, https://doi.org/10.5194/acp-15-3277-2015, 2015.

Zhang, Z., Xu, X., Qiao, L., Gong, D., Kim, S-J., Wang, Y., and Mao, R.: Numerical simulations of the effects of regional topography on haze pollution in Beijing, Sci. Rep., 8, 5504, https://doi.org/10.1038/s41598-018-23880-8, 2018.

Zheng, G. J., Duan, F. K., Su, H., Ma, Y. L., Cheng, Y., Zheng, B., Zhang, Q., Huang, T., Kimoto, T., Chang, D., Pöschl, U., Cheng, Y. F., and He, K. B.: Exploring the severe winter haze in Beijing: the impact of synoptic weather, regional transport and heterogeneous reactions, Atmos. Chem. Phys., 15, 2969–2983, https://doi.org/10.5194/acp-15-2969-2015, 2015.

Zhong, J., Zhang, X., Wang, Y., Sun, J., Zhang, Y., Wang, J., Tan, K., Shen, X., Che, H., Zhang, L., Zhang, Z., Qi, X., Zhao, H., Ren, S., and Li, Y.: Relative contributions of boundary-layer meteorological factors to the explosive growth of $PM_{2.5}$ during the red-alert heavy pollution episodes in Beijing in December 2016, J. Meteorol. Res., 31, 809–819, https://doi.org/10.1007/s13351-017-7088-0, 2017.

---

## Author Response (AR2)

Dear Editor,

Thank you for your efficient work in processing our manuscript entitled "Superimposed effects of typical local circulations driven by mountainous topography and aerosol-radiation interaction on heavy haze in the Beijing-Tianjin-Hebei central and southern plains in winter" (egusphere-2022-780). We have carefully read the reviewers' comments. Based on your suggestions, we considered all reviewers' comments, and provided convincing and clear responses to the negative comments of Reviewer 1. Furthermore, we have carefully proof-read and revised the manuscript to minimize typographical, grammatical, and bibliographical errors. We sincerely hope the correction will meet with approval. All changes are marked in red in the revised manuscript. The point-by-point responses (in blue) to the comments (in bold) are as following.

Hong Wang

Yue Peng

**Authors' Response to Referee #1**

**The paper was revised by the authors. However, the main inconveniences are a strong obstacle to the publication of this paper. The first one is the extension of the episodes analysed, which is extremely short. And the second inconvenience is that most of the paper lies on modelling studies without the comparison with experimental measurements. The readers want to know the goodness of the agreement between the meteorological variables modelled and those observed. Without this comparison, we should assume that the calculated atmospheric flow is true, and that the model is a perfect representation of the real world. The opposite thing would be if a complex experimental setup would be used to investigate the atmospheric flow in a short time period. In this case, the results would indicate the real flow, not the modelled one. Consequently, in this reviewer's opinion this paper should not be published in Atmospheric Chemistry and Physics.**

**Response:** We thank the reviewer for taking the time to review our manuscript. However, we couldn't agree with the reviewer's rejection opinion after we have revised so much focusing on the comments proposed by the reviewer last time.

We put a lot of effort into improving the manuscript to meet the requirements of Atmospheric Chemistry and Physics, including extending the study period, adding content and figures, polished the full text, etc. Every comment from the reviewer was

considered seriously and the manuscript was modified carefully (in Reply on RC1). We are so sorry that the reviewer couldn't understand our large work to response to his comments. Therefore, focusing on the reason for rejection, we want to argue as following:

First, what we study is to reveal the interaction between pollution and circulation at the local scale, rather than at the climate or regional scale. Averaging results over a longer period may weaken the features of local scale. Furthermore, considering the reviewer's comment, we have extended 3 days to 14 days including three pollution periods and obtained the consistent results with the previous ones, which further supported our conclusions; Secondly, the ground-level and the valuable sounding observation data were used to compare with the simulation results for $PM_{2.5}$ and meteorological condition, the good evaluation results laid a solid foundation for the later simulation analysis. We achieved a qualitative and quantitative description of the local circulations based on the sensitive experiments.

In summary, we believe this study is of extreme merit and publishable in the journal.

**Authors' Response to Referee #2**

**General comments:**

**The authors of the manuscript entitled: Superimposed effects of typical local circulations driven by mountainous topography and aerosol-radiation interaction on heavy haze in the Beijing-Tianjin-Hebei central and southern plains in winter, have successfully responded to all my comments and made the necessary changes in their work. Therefore in my opinion this very interesting manuscript, in which clearly the authors have put a lot of effort, deserves to be published in ACP after minor revisions described in the specific comments section of my review.**

**Response:** We appreciate the reviewer for his or her recognition of our work. We revised the manuscript according to the valuable comments. Below is the point-by-point response.

**Specific comments:**

**1. Line 39: Please define the abbreviation PBL.**

**Response:** We revised it in Lines 39-40 of the revised manuscript.

**2. Line 71: Please check syntax.**

**Response:** We checked the syntax of this paragraph and revised it in Lines 70-74 of the revised manuscript.

**3. Line 109: Please define the type of r.**

**Response:** The "correlation coefficient (r)" was replaced with "Pearson correlation coefficient (R)" in Line 111 of the revised manuscript.

**4. Line 129-130: Please correct "green start" to "green stars" and "station" to "stations".**

**Response:** We corrected them in Line 132 of the revised manuscript.

**5. Figure S2: In the legend correct "c: black arrow" to "b: black arrow".**

**Response:** The error has been corrected in Figure S2 of the Supplement.